# Predictors of Sexual Function and Performance in Young- and Middle-Old Women

**DOI:** 10.3390/ijerph19074207

**Published:** 2022-04-01

**Authors:** Krzysztof Nowosielski

**Affiliations:** Department of Gynecology, Obstetrics and Gynecological Oncology, University Clinical Hospital, Medical University of Silesia, 40-752 Katowice, Poland; knowosielski@sum.edu.pl; Tel.: +48-502-027-943

**Keywords:** sexuality, sexual function, female, sexual dysfunction, body image, sexual excitation/inhibition

## Abstract

Background: Maintaining sexual life in older women might be perceived as a measure of healthy and successful aging. This study aimed to establish the prevalence of female sexual dysfunction (FSD) based on the DSM-5 criteria and predictors of sexual performance in older women. Methods: A total of 185 women between 55 and 79 years old were included in the study. Validated scales were used to assess sexual function (Changes in Sexual Function Questionnaire (CSFQ)), the propensity for sexual excitation/inhibition (Sexual Excitation/Sexual Inhibition Inventory for Women), and sexual anxiety/avoidance (Body Exposure during Sexual Activity Questionnaire (BESAQ)). Multiple regression was used to assess the factors affecting sexual function and the prevalence of FDS. Results: Sexual distress was diagnosed in 14.1% of women, and FSD in 16.2% (*n* = 53), whereas sexual problems (CSFQ) were diagnosed in 33.3% of women, and distressing sexual concerns in 35.1%. Women with a lower number of male sexual partners (β = −0.22), a better attitude toward sex (β = 0.25), higher importance of sex (β = 0.31), a higher level of sexual excitation (β = 0.34), and that were sexually active (β = 0.39) had better sexual function, as evaluated by CSFQ. Conclusions: Most older women remain sexually active. The diversity of sexual activity in later life should be affirmed to encourage women to be sexually active and to strengthen the response to sexual stimuli.

## 1. Introduction

The sexual health of older women may be discussed from two distinct points of view—as a spectrum of physiological changes in the life cycle, leading to possible shifts in the type and frequency of sexual behaviors, or as a period with an increased frequency of sexual problems due to a variety of reasons [1]. It must be underlined that for the majority of women, sex remains an important part of life [2], irrespective of being in a relationship [3]. Many studies focused on sexual activity define it as penile-vaginal intercourse, whereas some older women may concentrate rather on masturbation, cuddling, touching, or oral sex, rather than restrain from sexual activities [4]. It seems that a new sexual script in older women must be developed to comply with comorbidities or the increase with age in the need for intimacy with a partner [5].

A recent study by Gore-Gorzawska showed that up to 75% of older women may become sexually inactive due to different reasons, such as the lack of a partner or sexual dysfunction in a partner. However, in a population of Polish women, three main patterns emerged: “I am glad that sex does not concern me anymore”, “I am satisfied with my memories”, and “The right one or no one”. That pattern seems to be shaped by cultural sexual scripts, based on a strong belief in the ancillary role of women in society, where faithfulness to one partner is a must and happiness is possible only within the marriage. Such scripts are still alive in the older population, especially those with a strong commitment to the Church, which determines more traditional sexual roles and behaviors [4].

Maintaining a sexual life in older women might be perceived as an indicator of healthy and successful aging [6]. Sexual activity might be a reassurance that one’s partner has “witnessed their life, fulfilled their needs, and emotionally and socially anchored them in later life” [6]. Sexual activity in this context is perceived as a source of pleasure, with the feeling of youthfulness and vigor. Besides sexual enjoyment, social activity and general health are also strong predictors of successful aging [7].

Five dimensions of sexual heath in older women may be identified: physical intimacy, emotional closeness during sex, sexual compatibility, sexual satisfaction, and distress related to sexual function problems [2]. Similarly, factors influencing sexuality in this age group are believed to be psychological (including body image and anxiety), cultural (including religiosity and religion), biological (testosterone level and estrogen decline) and partner-related (having a partner, relationship quality, satisfaction, communication, intimacy, and attitudes toward sex) [8].

As the results of studies assessing sexual function in older women are inconclusive due to methodological biases, we aimed to use a validated set of criteria and instrument to assess different aspects of sexual activity in women over 55 years of age. Thus, the aim of this study was to establish the prevalence of female sexual dysfunction (FSD) and distress based on DSM-5 criteria, to evaluate the frequency of sexual problems and distress based on the Changes in Sexual Function Questionnaire (CSFQ)/Female Sexual Function Index (FSFI) and Female Sexual Distress Scale (FSDS), respectively, and finally to assess the factors influencing sexual function and the prevalence of FSD in order to propose recommendations for health care professionals regarding how to enhance sexual performance in this group of women.

## 2. Materials and Methods

### 2.1. Participants

A total of 432 women over 55 years old were eligible for this cross-sectional population-based study conducted between 1 January 2021 and 31 December 2021. The individuals were recruited from women attending a gynecological outpatient clinic for a routine gynecological yearly visit and through a web-based advertisement over social media (Facebook). No identification data were required. However, all participants had to mark the “Yes” answer to agree to participate in this study.

The inclusion criterion was agreeing to participate in the study. Women currently diagnosed with cancer or being treated due to oncological disorders; women with severe illnesses disabling sexual activity or with severe psychological disorders; women who had experienced myocardial infarction (MI) less than 6 weeks prior; and women not willing to participate in the project were excluded from the study.

### 2.2. Procedure

All eligible women were asked to fill in the research questionnaire, available in paper-pencil form or online, at www.surveymonkey.pl (accessed on 1 January 2021). All participants had to sign an informed consent form in the case of the paper-pencil version, or click “YES” to agree to participate in the study in the online version. Out of these women, 213 did not agree to participate (the response rate was 53.2%), 22 had a history of psychiatric disorders, 2 were currently less than 6 months post MI, 10 underwent recent oncological treatment, and 11 returned incomplete questionnaires. Finally, 185 individuals were included in the final analysis (Figure 1).

### 2.3. Measurements

Young-old and middle-old women were those between 60–69 and 70–79, respectively [1].

Sexual activity was defined as vaginal, oral, or anal sex, masturbation, or mutual masturbation.

Risky sexual behaviors (RBS) was defined as “sexual contact with more than one sexual partner at the same time, engaging in sexual activity with a casual person (one-night stand), frequent change of sexual partners, having intercourse with a person living with HIV, inconsistent use of condoms in oral, anal, and vaginal contact except within the current relationship, prostitution or using the services of an escort agency, sexual contact under the influence of psychoactive substances other than alcohol and marijuana (chemsex), and drug injection with shared needles within the last 6 months” [9].

The questionnaire used in the study contained general medical history questions, demographic and socio-economic questions, and the battery of standardized and validated scales. CSFQ and FSFI were used to assess sexual function—scoring ≤41 and ≤27.5 points, respectively, was indicative of sexual problems [10]. FSDS-R was used to evaluate distress with a cut-off score ≥13 [11]. Based on the Diagnostic and Statistical Manual of Mental Disorders, 5th Edition (DSM-5) criteria, five questions were used to assess FSD [12]. HADS were used for assessing the general anxiety level and depressive symptoms. Propensity for sexual excitation/inhibition was assessed by the Sexual Excitation/Sexual Inhibition Inventory for Women (SESIIW) [12]. The Body Exposure during Sexual Activity Questionnaire (BESAQ) was used to assess body image and sexual avoidance [13]. Sexual satisfaction, attitudes toward sex, the importance of sex, relationship quality, satisfaction with a partner as a lover, and sexual life quality were assessed by 5-point Likert scale-based questions. The severity of menopausal symptoms was assessed by the Menopausal Rating Scale (MRS) [9]. Finally, sexual self-schemas were evaluated by the Sexual Self-Schema Scale [14]. All the scales used in the project are described elsewhere [9,10,12,13,15].

The study protocol was approved by the Ethical Committee of the Silesian Chamber of Physicians and Dentists in Katowice, Poland (decision number—SIL:/KB/756p/15).

### 2.4. Study Sample Calculation

According to statistical books in Poland from 2021, 3,279,161 women were between 60 and 70 years old. A minimum sample of 183 individuals was required, with a 95% confidence interval (CI) and 7% margin of error. Additionally, assuming that the prevalence of FSD based on the DSM-5 criteria in Poland would be 15% [9], the minimum sample size required would be 181 with a CI of 95% and a margin of error of 5%. As at least 20% of the sample is typically lost (returning empty questionnaires), the initial sample should be at least 219 women.

### 2.5. Statistical Analysis

Statistical analysis was performed in Statistica 12.0 for Windows (StatSoft, Warsaw, Poland). Missing values were assessed for all variables (less than 5%). Skewness and kurtosis were assessed to check for univariate and multivariate distribution normality. Values larger than 3 for skewness or larger than 10 for kurtosis were considered to indicate nonnormality [12]. To assess factors influencing sexual function (based on CSFQ) and the frequency of sexual dysfunction (DSM-5 criteria), univariate linear and logistic regression models were used. In the first step, all variables were checked for a significant contribution to the assessed parameters. In the final step, only those statistically significant in the first step were introduced to the model—multivariate forward stepwise regression. The final models were established. Additionally, the chi-squared test and Mann-Whitney U test were used to compare women with and without FSD, sexual problems, and distressing sexual concerns. *p*-values less than 0.05 were considered statistically significant.

## 3. Results

The mean age of the investigated population was 67.8 ± 3.08 (56.0–79.4) years. A total of 87.6% (*n* = 162) were in a relationship, and 81.1% (*n* = 150) were sexually active in the last 12 weeks (Table 1 and Table 2). A total of 33.5% (*n* = 62) and 38.4% (*n* = 71) scored above the threshold for depressive symptoms and anxiety in HADS, respectively. A total of 26.5% of women had arterial hypertension, 3.2% had diabetes mellitus, 36.7% were using medications for any reason, and 35.1% (65) were using menopausal hormonal therapy (MHT).

The evaluation of sexual function revealed that sexual satisfaction was rated as high (3.6 points), with a high importance of sex and the quality of the relationship (3.3 and 4.1, respectively). Both the woman’s attitude and her partner’s attitude toward sex were positive (4.1 and 4.2 points, respectively). Sexual dysfunction was reported in 27% of partners. However, satisfaction from a partner as a lover was high (3.9 points). Propensity for sexual excitation and inhibition was 2.55 and 2.57 points, respectively, and the BESAQ score was 1.4.

Sexual distress was diagnosed in 14.1% (*n* = 26) and 22.2% (*n* = 41) of women based on the FSDS-R and DSM-5 criteria, respectively. FSD was noted in 16.2% (*n* = 53) of women, whereas sexual problems based on CSFQ were noted in 33.3% (*n* = 53) and distressing sexual concerns (based on FSFI and FSDS-R cut-offs) in 35.1% (*n* = 65). Most of the respondents were either schematic positive or co-schematic (43.3% and 47.7%, respectively) (Table 1 and Table 2).

Although a high frequency of sexual dysfunction in partners was noted, there was a difference neither in the frequency of sexual activity nor in the sexual function assessed by the CSFQ in women whose partner had or did not have sexual dysfunction.

The assessment of the subjective evaluation of current sexual function compared with that 10 years ago showed no change or slightly decreased in all measures of sexual function (Figure 2).

The between-group comparison (FSD vs. no FSD, sexual problems vs. no sexual problems, and distressing sexual concerns vs. no concerns) showed some statistical differences, as presented in Table 3. Additionally, the analysis of the qualitative variable revealed that the prevalence of FSD was statistically higher in women with primary education—45.4% had FSD compared with 12.5% and 20.0% in the secondary and higher education groups (*p* = 0.04). FSD was also more frequent in those not taking MHT (23.1% vs. 9.7%, *p* = 0.02). Distressing sexual concerns were more frequent in Roman Catholics compared with the followers of other religions and atheists (65% vs. 25% vs. 10%, respectively, *p* = 0.02). Finally, the prevalence of sexual problems (CSFQ scores) was lower in those consuming alcohol compared with non-consumers (30% vs. 61.5%, *p* = 0.02) and less frequent in those watching erotic videos (44.6% vs. 64.1%, *p* = 0.01).

The logistic regression model revealed a higher risk of FSD in those with a higher intensity of the psychological symptoms of menopause (OR = 1.37; CI: 1.2–1.6; *p* = 0.001), a lower propensity for sexual excitation (OR = 0.37; CI: 0.1–0.9; *p* = 0.04), and with primary education compared with secondary (OR = 0.07, CI: 0.01–0.4; *p* = 0.001) and higher education (OR = 0.1; CI: 0.02–0.7; *p* = 0.01).

In the multiple stepwise forward model, those with a lower number of lifetime male sexual partners (β = −0.22; t = −2.4, *p* = 0.02), a better attitude toward sex (β = 0.25, t = 2.89, *p* = 0.010), a higher importance of sex (β = 0.31; t = 7.2; *p* = 0.0010), a higher propensity for sexual excitation assessed by SESIIW (β = 0.34; t = 3.47; *p* = 0.001), and those that were sexually active in the last 3 months (β = 0.39, t = 4.6; *p* = 0.001) had better sexual function, as evaluated by CSFQ. The model showed excellent parameters (corrected R2 = 0.77, F = 27.8, *p* = 0.001).

## 4. Discussion

The purpose of the study was to assess sexual function, sexual performance, and the prevalence of FSD based on strict DSM-5 criteria in the population of young- and middle-old women. Furthermore, the study aimed to established correlates of sexual function and performance, including, for the first time, sexual self-schema, body image, and propensity for sexual excitation/inhibition, as well as risk factors for FSD in this population. It is believed that the holistic approach to sexuality presented in this paper enabled us to picture the interaction between the aforementioned factors and sexual health in the context of aging.

### 4.1. Sexual Activity and Relationship

The proportion of sexually active older women varies between studies; according to a recent paper, between 31% and 57.8% of women over 70 years are sexually active in the UK [8,16,17], between 35.7% and 42% in Australia, [18,19], and 61–75% in Belgium, Norway, Denmark, and Portugal [20]. However, in the current study, 81% of women reported being sexually active, which is in line with the latest results from a small observational study in Poland—68% [21]. It must be noted that the differences might be caused by various definitions of sexual activity used in the mentioned studies. Similarly, a high number of women being in a relationship was noted, in line with other observations, reporting figures from 52.6% (recent Australian cohort study) [18] to 87% (Belgium) [8,17,20,22]. According to a recent Polish study, 78% of women were in a relationship [21].

### 4.2. Sexual Performance and Satisfaction

The results of the current study show that the importance of sex and sexual satisfaction was in the upper limits (3.3 and 3.6, respectively). In a recent study by Smith et al. sexual enjoyment was shown to be associated with better well-being, and 87% of participants were satisfied with their sexual life [17]. The results from four European countries demonstrated, however, a decrease in enjoyment of sex compared with that 10 years earlier [2]. Additionally, women over 55 years reported a lower satisfaction with their sex life, but a higher importance of sex compared with those under 55. The main reason to have sex was for pleasure and was similar throughout the life cycle [22]. Similar results were illustrated in a German sample—the mean importance of sex was 4.2 points [23], whereas in a recent Polish paper, satisfaction from sexual life was assessed to be 3.3 points and to decrease with age [23].

### 4.3. Body Image and Sexual Excitation/Inhibition

Body image evaluation showed a higher level of sexual anxiety and avoidance according to BESQA compared with the general Polish population (1.4 vs. 1.25 for the general Polish population and 1.11 for the USA sample [13]). Similarly, the sexual excitation score was lower compared with that in the general population (SES-2.55 and SIS-2.63) [12], showing a decreasing propensity for sexual excitation with a stable sexual inhibitory tone.

### 4.4. Sexual Distress, Problems and Dysfunction

In the current study, sexual distress was seen in 14.1% and 22.2% of women (DSM-5 and FSDS-R, respectively), FSD in 16.2% (DSM-5), sexual problems (CSFQ) in 33.3%, and distressing sexual problems (FSFI + FSDS-R) in 35.1%. According to the Female Sexual Problems Associated with Distress and Determinants of Treatment Seeking (PRESIDE) study, up to 44.2% of women report sexual concerns, whereas up to 22.8% experience sexual distress, and 12% experience distressing sexual problems. Although the prevalence of sexual problems increases with age (from 44.6% between 45 and 64 years to 80.1% over 65 years), the distress drops from 12.3% to 7.4%, respectively. Arousal problems were the most dominant in an Australian cohort of women, affecting 13.6% of women over 65 years of age [3]. In contrast, Zeleke et al. showed that the frequency of low desire assessed by FSFI is up to 90% of women, whereas the prevalence of any sexual problem was seen in 6% to 25% of respondents [18]. Similarly, Nazarpoir et al. in their metanalysis, reported that up to 85% of older women complained of sexual problems [24]. Based on FSFI scores, sexual problems were reported by 62% of Turkish women [25]. However, in another study, sexual concerns were seen in less than 10% of women, difficulties in achieving orgasm were seen in 26.4%, and low arousal was seen in 33.1% [17]. When FSFI and FSDS scores were combined, distressing sexual concerns (defined as epidemiological hypoactive sexual desire disorder) was present in 13.6% of respondents in a recent Australian study [18], and in 33% between 55 and 60 years, with a drop to 12.9% in the 70–74 years interval [19]. The distress was reported to be present in 15.5% of Australian older women [18]: in 44% between 50 and 55 years old and in 15% of those 70 years or older [19]. Surprisingly, the frequency of FSD is this study is not higher compared with that in women of perimenopausal age in Poland—14.7% [9]. This might reflect the increase in the prevalence of sexual problems (30% in perimenopausal women vs. 33% in current study) with a decline in distress (19.5% in perimenopause vs. 14.1% in current study) [9].

### 4.5. Risk Factors for FSD

In the current study, it was noted that FDS is associated with the psychological symptoms of menopause and attitudes toward sex, but not with depressive symptoms. Some correlates of being sexually active were noted in previous studies: past sexual experiences, partner’s illnesses, partner’s interest (attitude) in sex, positive sexual schema (more romantic-passionate), importance of sex, smoking, positive attitudes toward sex, having a partner, and self-esteem. BMI, religious commitment, church attendance, and comorbidities showed mixed results [1]. Vulvovaginal symptoms of menopause are acknowledged by 62% of women (moderate to severe symptoms), and those symptoms are believed to be associated with worse sexual function [3]. Similarly, depression was associated with FSD [3].

### 4.6. Predictors of Sexual Function and Performance

Some factors were shown to be associated with sexual function and satisfaction: a higher number of sexual contracts contributed to a higher global life satisfaction [21], having a partner (OR = 5.4–7.9), higher education (OR = 2.6–5.9), higher importance of sex (β = 0.08), absence of depression (β = −0.16), and lower education compared with medium (β = −0.08) or higher education (β = −0.11) [20,23]. Being unpartnered (OR = 1.6–4.2), vaginal dryness (OR = 2.37), depressive symptoms (OR = 4.15) [18], lower education, and being sexually inactive (5.26) [19] were associated with worse sexual function and risk of FSD. It was also shown that sexual inactivity and lower education were related to worse sexual function. This is in line with the results of some recent studies [20]. However, in contrast to some observations by Gore-Gorzawska [4], no associations between church attendance and religiosity and sexuality were observed.

### 4.7. Final Remarks on Sexual Function

The sexual function of young- and middle-old women in Poland seems to be sufficient, and not notably different from observations in other nations. This is in contrast with recent papers showing some cultural differences among nations, such as in Norwegian couples, where “male partner’s emotional intimacy contributed to his female partner’s sexual well-being through more frequent sex”. It cannot be excluded that gender roles in Polish women rooted in the post-Soviet paternalistic cult of man, where a woman must prioritize sexually satisfying her man over her own needs [26], may shape sexual performance (concentration on partner’s desire). However, in general, the tendency is similar; the higher the intimacy and frequency of sexual acts, the greater the sexual function [2,20]. It might be speculated that all older women are similar and that expected cultural differences (especially in Eastern post-Soviet countries) are no longer critical. Other factors, such as propensity for sexual excitation and being sexually active, may play a dominant role.

### 4.8. Implications for Clinical Practice and Recommendations

Sexual function and the prevalence of FSD depend on the importance of sex, attitudes toward sex, sexual activity level, and propensity for sexual cues, which are all modifiable. Health care professionals counseling older women due to medical reasons should address those factors and introduce educational interventions to enhance sexual function and performance in that group of women. Furthermore, the diversity of sexual activity in later life should be also affirmed in order to encourage women to be sexually active [27] and to strengthen their response to sexual stimuli.

### 4.9. Study Limitations

This study has some limitations. The study sample might not be large enough to generalize the obtained results to the general population of women over 55 years old. Secondly, sexual satisfaction was assessed by a single question; however, a similar methodology has been widely used [28]. Thirdly, we did not concentrate on sexual activity per se, as in the recent metanalysis by Bell et al. [1]. Fourthly, gender roles rooted in the paternalistic cult of men, where a woman is guided more by the satisfaction of the partner than by her own desire, could have biased the obtained results. Finally, partner sexual function was not assessed, which could give another perspective but would limit the subjects to only those in relationship. Despite the aforementioned limitations, the study is believed to be of good quality, adding important information to the current knowledge on sexual health in young and middle-old women. However, further studies on a larger group of women are required to confirm the findings of this particular study.

## 5. Conclusions

Most young- and middle-old women remain sexually active, generally with good sexual function, and a low level of distress and FSD. However, levels of sexual anxiety and avoidance are high.

## Figures and Tables

**Figure 1 ijerph-19-04207-f001:**
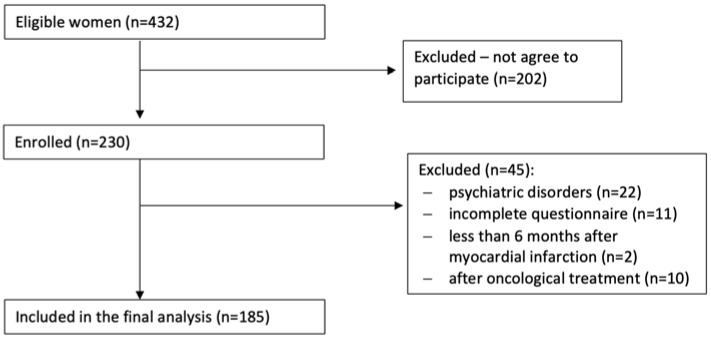
Study flow chart.

**Figure 2 ijerph-19-04207-f002:**
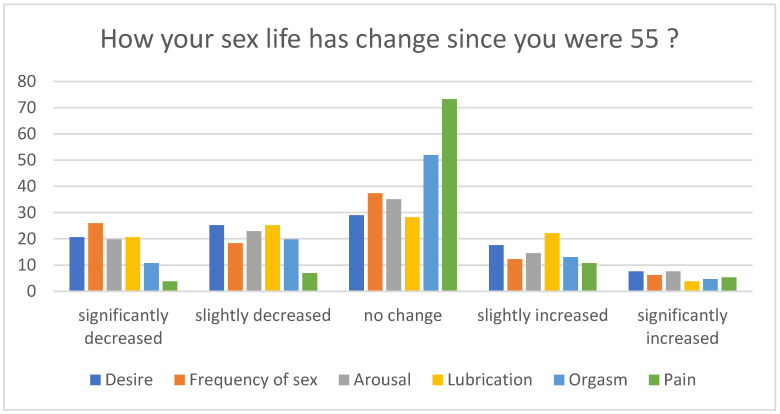
Changes in sexual function according to respondents’ opinion.

**Table 1 ijerph-19-04207-t001:** General characteristic of the studies population—descriptive statistics.

Variable	% (*n*)
Residency	
Urban	84.3 (156)
Rural	15.7 (29)
Education	
Primary	7.0 (13)
Secondary	45.9 (85)
Higher	47.3 (87)
Education	
Black-collar	22.2 (41)
White-collar	62.7 (116)
Unemployed	7.0 (13)
Retired	8.1 (15)
Smoking (Yes)	23.2 (43)
Drugs (Yes)	1.6 (3)
Alcohol (Yes)	13.5 (25)
Religion	
Catholic	76.8 (142)
Other	7.0 (13)
Atheist	16.2 (30)
Church attendances	
More than 1 time a week	20.0 (37)
Once a month	10.8 (20)
One or two times a year	43.1 (80)
Never	26.1 (48)
Participation in religious practices	37.2 (69)
Being in RS (Yes)	87.6 (162)
Sexual activity in last 3 months	81.1 (150)
Sexual dysfunction in partner	27.0 (50)
Watching erotic videos (Yes)	39.5 (73)
Sexual abuse in childhood (Yes)	9.2 (17)
Sexual behaviors	
WSW	91.9 (170)
WSMW	3.8 (7)
WSM	2.2 (4)
Asexual	2.2 (4)
Sexual orientation	
Heterosexual	92.9 (172)
Homosexual	2.7 (5)
Bisexual	3.2 (6)
Asexual	1.1 (2)
RSB (Yes)	10.8 (20)
Regularity of menstruation (Yes)	64.8 (120)
Pregnancies (Yes)	90.8 (168)
MHT (Yes)	35.1 (65)
HADS-Depression (Yes)	33.5 (62)
HADS-Anxiety (Yes)	38.4 (71)
Distress (FSDS-R)	14.1 (26)
Distress (DSM-5)	22.2 (41)
FSD (FSFI + FSDS-R scores)	35.1 (65)
FSIAD	13.0 (24)
FOD	10.3 (19)
GPPPD	4.9 (9)
FSD (DSM-5)	16.2 (30)
Sexual problems—CSFQ	33.3 (53)
Pleasure—CSFQ	77.3 (143)
Desire/Frequency—CSFQ	64.3 (119)
Desire/Interest—CSFQ	64.3 (119)
Arousal/Excitement—CSFQ	78.9 (146)
Orgasm/Completion—CSFQ	49.1 (91)
Sexual self-Schema—positive	43.3 (80)
Sexual self-Schema—negative	4.5 (8)
Sexual self-Schema—Aschematic	5.5 (8)
Sexual self-Schema—Co-schematic	47.7 (88)

RS—relationship, RSB—Risky Sexual Behaviors, WSW—woman who has sex with woman; WSM—woman who has sex with man, WSWM—woman who has sex with woman and man; MHT—Menopause Hormonal Therapy, CSFQ—Changes in Sexual Function Questionnaire, FSD—Female Sexual Dysfunction, FSFSI—Female Sexual function Index, FSDS-R—Female Sexual distress Index-Revised, FISAD—Female Sexual Interest/Arousal Disorder, FOD—Female Orgasmic Disorder, GPPPD—Genito-Pelvic Pain/Penetration Disorder, HADS—Hospital Anxiety and Depression Scale.

**Table 2 ijerph-19-04207-t002:** General characteristic of the studied population—quantitative variables.

Variable	Mean	Minimum	Maximum	SD
Age	67.82	56.00	79.37	3.08
Religiosity	2.93	1.00	5.00	1.13
Nr of cigarettes a day	2.12	0.00	30.00	4.89
BMI	26.51	13.74	41.15	4.62
Weight self-perception	3.59	1.00	5.00	0.74
Age of first genital sex	19.27	14.00	30.00	2.66
Age of first oral sex	21.37	13.00	40.00	5.54
Age of first masturbation	17.68	10.00	45.00	5.65
Importance of sex	3.31	1.00	5.00	0.83
Satisfaction from sex life	3.68	1.00	5.00	0.91
Attitude toward sex	4.11	1.00	5.00	0.76
Duration of RS	21.98	0.00	45.00	12.41
Quality of RS	4.12	1.00	6.00	1.14
Satisfaction from a partner as a lover	3.96	1.00	6.00	1.20
Partner’s attitude toward sex	4.19	1.00	6.00	0.92
Vaginal sex/month	6.27	0.00	100.00	8.97
Cuddling/month	4.90	0.00	40.00	6.78
Anal sex/month	0.68	0.00	15.00	2.13
Oral sex/month	2.26	0.00	20.00	4.07
Mutual masturbation/month	2.18	0.00	25.00	4.45
Self-masturbation/month	1.68	0.00	30.00	4.72
Orgasm/month	5.72	0.00	50.00	7.45
Sex events/month	6.03	0.00	100.00	9.72
Sexual satisfaction/month	5.95	0.00	90.00	8.75
Nr of lifetime male sexual partners	4.09	0.00	33.00	4.38
Nr of lifetime female sexual partners	0.49	0.00	20.00	2.53
Nr of deliveries	1.85	0.00	6.00	0.92
HADS—Depression	8.51	0.00	25.00	6.02
HADS—Anxiety	10.66	0.00	26.00	7.02
FSFI—Desire	4.03	1.20	6.00	1.28
FSFI—Arousal	3.42	1.20	7.20	1.56
FSFS—Lubrication	4.15	1.20	7.20	1.59
FSFI—Orgasm	3.68	1.20	7.20	1.47
FSFI—Satisfaction	2.72	1.20	6.00	1.24
FSFI—Pain	4.22	1.20	6.40	1.48
FSFI—total score	22.23	10.20	39.00	5.76
FSDS-R—total score	14.67	0.00	52.00	13.02
BESAQ	1.40	0.04	4.00	0.68
CSFQ—pleasure	3.30	1.00	5.00	1.08
CSFQ—desire/frequency	5.86	2.00	10.00	1.58
CSFQ—desire/interest	7.56	3.00	15.00	2.51
CSFQ—arousal/excitement	9.48	3.00	15.00	2.50
CSFQ—orgasm/completion	10.69	3.00	15.00	2.53
CSFQ—total score	43.71	15.00	69.00	8.87
SES	2.55	1.00	3.88	0.58
SIS	2.57	1.00	4.00	0.50
WMRQ—Intimacy	27.92	8.00	40.00	7.59
WMRQ—Disappointment	34.65	10.00	50.00	9.32
WMRQ—Self-realization	24.90	7.00	35.00	6.06
WMRQ—Similarity	25.03	7.00	35.00	6.15
WMRQ—total score	103.12	57.00	160.00	14.62
MRS—total	14.45	0.00	44.00	8.67
MRS—psychological domain	5.42	0.00	16.00	3.90
MRS—somatic domain	5.33	0.00	16.00	3.41
MRS—urological domain	3.71	0.00	12.00	2.83

SD—standard deviation; BMI—Body Mass Index, RS—relationship, FSFS—Female Sexual Function Index, FSDS-R—Female Sexual Distress Scale-Revised, SES—Sexual Excitation Scale, SIS—Sexual Inhibition Scale, WMRQ—Well-Matched Relationship Questionnaire, MRS—Menopausal Rating Scale, CSFQ—Changes in Sexual Function Questionnaire, BESAQ—Body Exposure during Sexual Activity Questionnaire.

**Table 3 ijerph-19-04207-t003:** Comparison between different group of studied women according to the presence of sexual dysfunctions, concerns, and problems.

Variable (Mean)	FSD	No FSD	*p*	Distressing Sexual Concerns	No Concerns	*p*	Sexual Problems	No Sexual Problems	*p*
Importance of sex	2.90	3.46	0.00	3.00	3.57	0.02	3.08	3.74	0.00
Satisfaction from sex life	3.20	3.88	0.00	2.75	4.00	0.00	3.47	4.09	0.00
Quality of RS	3.67	4.26	0.02	3.90	4.49	0.19	4.03	4.32	0.02
Satisfaction from a partner as a lover	3.37	4.10	0.01	3.10	4.41	0.00	3.70	4.37	0.01
Partner’s attitude toward sex	3.73	4.24	0.03	4.05	4.54	0.03	4.02	4.35	0.03
Attitude towards sex	3.63	4.25	0.00	4.00	4.41	0.10	3.87	4.50	0.00
Vaginal sex/month	3.96	6.53	0.00	3.67	6.56	0.02	5.35	7.22	0.00
Anal sex/month	0.36	0.73	0.48	0.00	0.56	0.04	0.61	0.77	0.48
Oral sex/month	1.00	2.41	0.08	1.28	3.75	0.02	1.70	2.54	0.08
Mutual masturbation/month	1.89	2.04	0.64	2.11	4.11	0.05	1.62	2.58	0.64
Self-masturbation/month	2.68	1.27	0.62	6.00	1.26	0.75	1.52	1.50	0.62
Orgasm/month	3.29	6.09	0.00	8.61	7.50	0.73	4.03	7.58	0.00
Sex events/month	3.21	6.07	0.00	8.89	8.71	0.69	4.13	6.70	0.00
Sexual satisfaction/month	3.04	6.38	0.00	3.39	7.61	0.01	4.20	7.78	0.00
Nr of lifetime male sexual partners	2.86	3.90	0.18	3.25	6.11	0.01	3.10	4.24	0.18
HADS—Depression	11.07	7.82	0.00	15.45	15.97	0.91	7.88	8.57	0.00
SES	2.34	2.59	0.05	2.50	2.93	0.00	2.35	2.80	0.05
SIS	2.58	2.56	0.97	2.75	2.69	0.45	2.50	2.66	0.97
WMRQ—Intimacy	26.04	28.28	0.13	22.07	26.86	0.04	26.48	29.81	0.13
WMRQ—Disappointment	30.00	35.55	0.01	28.93	33.04	0.09	33.04	36.79	0.01
WMRQ—Similarity	23.50	25.32	0.09	20.14	24.11	0.05	23.45	27.09	0.09
WMRQ—total score	104.35	102.89	0.43	92.77	100.04	0.16	101.19	105.67	0.43
MRS—total	22.15	12.89	0.00	15.27	16.10	1.00	15.57	12.97	0.00
MRS—psychological domain	8.85	4.72	0.00	6.00	6.66	0.81	5.78	4.92	0.00
MRS—somatic domain	7.81	4.83	0.00	5.60	5.55	0.71	5.65	4.91	0.00
MRS—urological domain	5.50	3.34	0.00	3.67	3.90	0.90	4.14	3.14	0.00

FSD—Female Sexual Dysfunction, RS—relationship, SES—Sexual Excitation Scale, SIS—Sexual Inhibition Scale, WMRQ—Well-Matched Relationship Questionnaire, MRS—Menopausal Rating Scale, CSFQ—Changes inn Sexual Function Questionnaire.

## Data Availability

The data that support the findings of this study are available upon request from the author (K.N.). The data are not publicly available due to ethical reasons.

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
