# Peer review of "Predictors of Sexual Function and Performance in Young- and Middle-Old Women"

_ijerph, 2022, doi:10.3390/ijerph19074207_

Round 1
Reviewer 1 Report
This is a study aiming to to assess different aspect of sexually in women over 55 years of age. The presentation of results is not clear enough.
Since the study sample was described in such great detail in Table 1 I think it would be interesting to evaluate comparatively the parameters assessed in at least some of the subgroups defined in Table 1, otherwise the details described in the table present no interest to the readers.
The abbreviation SD is confusingly used for both standard deviation and sexual dysfunction
Numerous English spelling/grammar mistakes are present throughout the manuscript
The figure mentioned in the manuscript was not available
Author Response
Dear Reviewer,
Thank you for tour valuable comments
The answers are written below.
1. This is a study aiming to assess different aspect of sexually in women over 55 years of age. The presentation of results is not clear enough.
Ad 1. The results section was read through and changed according to the reviewer’s suggestions
2. Since the study sample was described in such great detail in Table 1 I think it would be interesting to evaluate comparatively the parameters assessed in at least some of the subgroups defined in Table 1, otherwise the details described in the table present no interest to the readers.
Ad 2. As suggested a paragraph comparing women with FSD, sexual problems and distressing sexual problems was added.
3. The abbreviation SD is confusingly used for both standard deviation and sexual dysfunction
Ad 3. As suggested, the abbreviation SD (meaning sexual dysfunction) was changed to sexual dysfunction throughout the text
4. Numerous English spelling/grammar mistakes are present throughout the manuscript
Ad 4. The MS was corrected by English Editing services
5. The figure mentioned in the manuscript was not available
Ad 5. The figure was added to the MS.
Reviewer 2 Report
The authors analyze the prevalence of female sexual dysfunction (FSD) and distress based on DSM-5 criteria and predictors of sexual performance in older women. They applied validated questionnaires to 185 women between 55 and 79 years old. Applying regression models, the authors found sexual distress in 14.1% and FSD in 16.2 % of the participants. Other findings were: women with a lower number of male sexual partners, a better attitude toward sex, higher importance of sex, higher sexual excitation, and being sexually active had a better sexual function.
My comments are as follows:
1) The document has many typographical errors. Fix all cases.
2) The introduction and conclusion sections should be expanded.
3) Give more details of the applied questionnaires P.e. its validation metrics (Cronbach's alpha and those that apply), the number of items, and their domains (or aspects).
4) In the statement: “The assessment of subjective evaluation of current sexual function compared to that before 10 yeas showed no change or slightly decrease in all sexual function – Figure 1.” You refer to a figure that has no relation to what was said.
5) In paragraphs 205-210 values appear in which it is not indicated what they refer to, e.g. “In the multiple stepwise forward model those with lower nr of lifetime male sexual partners (=0.22; t=-2.4, p=0.02),…”. What does 0.22 mean? What does nr mean? Check and correct all cases in that paragraph and throughout the document.
6) Include the recommendations that you mention at the end of the introduction.
7) Future work should be mentioned.
Author Response
Dear Reviewer,
Thank you for your valuable remarks. The answers are written below.
The authors analyze the prevalence of female sexual dysfunction (FSD) and distress based on DSM-5 criteria and predictors of sexual performance in older women. They applied validated questionnaires to 185 women between 55 and 79 years old. Applying regression models, the authors found sexual distress in 14.1% and FSD in 16.2 % of the participants. Other findings were: women with a lower number of male sexual partners, a better attitude toward sex, higher importance of sex, higher sexual excitation, and being sexually active had a better sexual function.
My comments are as follows:
1) The document has many typographical errors. Fix all cases.
Ad 1 – corrected
2) The introduction and conclusion sections should be expanded.
Ad 2. Thank you for that remark. However, based on other reviews I decided not to increase that length of the text as it should be condense – it is not a review but original paper.
3) Give more details of the applied questionnaires P.e. its validation metrics (Cronbach's alpha and those that apply), the number of items, and their domains (or aspects).
Ad 3. Similarly, all used questionnaire are well recognized and for that reason no detailed description was provided in order to make the MS “easy to read”
4) In the statement: “The assessment of subjective evaluation of current sexual function compared to that before 10 yeas showed no change or slightly decrease in all sexual function – Figure 1.” You refer to a figure that has no relation to what was said.
Ad 4. The missing figure was added.
5) In paragraphs 205-210 values appear in which it is not indicated what they refer to, e.g. “In the multiple stepwise forward model those with lower nr of lifetime male sexual partners (=0.22; t=-2.4, p=0.02),…”. What does 0.22 mean? What does nr mean? Check and correct all cases in that paragraph and throughout the document.
Ad 5. Sorry for that mistake. It was supposed to be b. That was corrected as suggested.
6) Include the recommendations that you mention at the end of the introduction.
Ad 6. That recommendations were described in the paragraph 4.8. Implications for clinical practice
7) Future work should be mentioned.
Ad 7. Added as requested
Reviewer 3 Report
The authors have presented a well written and interesting paper on female sexual function. I have no further concerns with the present paper and I fully endorse it's publication, please check only some typing mistakes.
Author Response
Dear Reviewer,
Thank you for engaging in the review process.
As suggested all typing mistakes were corrected
Reviewer 4 Report
Two aspects of this study have not been clearly described:
It is not stated whether informed consent has been requested from the participants and how data protection has been carried out.
I consider that the gender role of Polish women has not been taken into account, which can exert an enormous influence on the maintenance of sexual relations guided more by the satisfaction of the partner than by the woman's own desire, and this can be a serious limitation that biases the results of the study.
Maintaining sexual life in older women might be perceived as a sing for 9
health and successful aging: This terminology is more appropriate for a popular rather than a scientific article.
Young-old and middle-old women were those between 60-69 and 70-79, respectively: Description is misleading, so it is necessary to use other terminology.
Based on the assumption a minima sample of 183 individuals was required, with 95%: This figure is representative of the country or an area?
Author Response
Dear Reviewer,
Thank you for your valuable comments. The response are written below.
Two aspects of this study have not been clearly described:
1. It is not stated whether informed consent has been requested from the participants and how data protection has been carried out.
Ad 1. All participants had to sing an informed consent form in case or paper-pencil version or click “YES” to agree to participate in the study in online version. That information was added to the material an method section\
2. I consider that the gender role of Polish women has not been taken into account, which can exert an enormous influence on the maintenance of sexual relations guided more by the satisfaction of the partner than by the woman's own desire, and this can be a serious limitation that biases the results of the study.
Ad 3. Thank you for that remark. I addressed that in the discussion and limitation of the study section.
3. Maintaining sexual life in older women might be perceived as a sing for 9
health and successful aging: This terminology is more appropriate for a popular rather than a scientific article.
Ad 3. Thank you for that remark. The sentence was changed to “Background. Maintaining sexual life in older women might be perceived as a surrogate of healthy and successful aging.”
4. Young-old and middle-old women were those between 60-69 and 70-79, respectively: Description is misleading, so it is necessary to use other terminology.
Ad 4. That term are commonly used in the scientific literature – please see ell, S.; Reissing, E.D.; Henry, L.A.; VanZuylen, H. Sexual Activity After 60: A Systematic Review of Associated Factors. Sexual Medicine Reviews 2017, 5, 52-80, doi:https://doi.org/10.1016/j.sxmr.2016.03.001.
5. Based on the assumption a minima sample of 183 individuals was required, with 95%: This figure is representative of the country or an area?
Ad 5. This is representative for the country.